# The influence of perceived medical risks and psychosocial concerns on photoprotection behaviours among adults with xeroderma pigmentosum: a qualitative interview study in the UK

Myfanwy Morgan,[1] Rebecca Anderson,[1] Jessica Walburn,[1] John Weinman,[1] Robert Sarkany[2]

¹School of Cancer and Pharmaceutical Sciences, King's College London, London, UK
²National Xeroderma Pigmentosum Service, Guy's and St Thomas' NHS Foundation Trust, London, UK

**Correspondence to**
Professor Myfanwy Morgan; myfanwy.morgan@kcl.ac.uk

## ABSTRACT

**Background** A high level of photoprotection is required by people with xeroderma pigmentosum (XP), a rare skin disease, to reduce skin cancer and other risks. However poor photoprotection is thought to be widespread.

**Purpose** This study examines the influences on photoprotection behaviours in adults with XP.

**Design** Inductive qualitative study with semistructured interviews. Analysis employed a framework approach.

**Setting** National sample recruited through a specialist XP centre in London.

**Methods** Semistructured interviews at patients' homes. All transcripts were coded and themes charted for each participant. Comparisons within and across cases identified common themes and differing motivations and approaches to photoprotection. Credibility of interpretations assessed through patient/carer input and clinic adherence scores.

**Participants** 25 adults (17 male, eight female) aged 16–63 years with diagnosed XP attending a specialist centre. 18 lived outside London.

**Results** Awareness of risks of ultraviolet radiation (UVR) and photoprotection was high. However, photoprotection behaviours varied according to perceived necessity and concerns. Three behavioural responses were identified: (1) '*dominated*' by planning and routines to achieve a high level of photoprotection with significant activity restrictions and psychosocial impacts. (2) '*resistant*' to photoprotection with priority given to avoiding an illness identity and enjoying a normal life. (3) 'Photoprotection' *integrated*' with an individual's life with little psychosocial impact. These responses were influenced by illness, personal and contextual factors including age, life stage and social support. Only the 'integrated' group achieved an equilibrium between perceived 'necessity' and 'concerns'.

**Conclusions** The personal balance between perceived risks of UVR and social/psychological 'concerns' led to differing behavioural responses and contributes to an understanding of adaptation and normalisation in chronic illness. The study will also inform a series of individualised behavioural interventions to reduce measured UVR exposure among people with XP that are potentially applicable to other conditions with high risks of skin cancer.

### Strengths and limitations of the study

► Qualitative home-based semistructured interviews elicited detailed accounts of individuals' perceptions and photoprotection behaviour.
► The analysis went beyond identification of individual themes to identify three multidimensional categories of response to photoprotection, with their credibility enhanced through discussion with the patient/carer (patient and public involvement) group and data on prior specialist nurse ratings of adherence.
► The research was conducted with patients in the National Health Service who attended a national specialist centre for xeroderma pigmentosum (XP) which may have influenced their knowledge and attitudes to XP.
► It was not feasible to conduct follow-up interviews to examine the influences on transitions between different modes of response given the significant demands of the overall research programme on a small sample of people with this rare disease.

## INTRODUCTION

Globally the prevalence of chronic disease is increasing but adherence to recommended therapies remains persistently low; on average 50% of all patients with long-term therapy for chronic disease in developed countries are non-adherent to medication[1] and 50%–80% to behavioural changes.[2] Both forms of non-adherence increase risks of morbidity and mortality with implications for demands on health services and costs of care.[3]

Research conducted to explain high levels of non-adherence with medical advice has identified a range of illness-related, psychosocial and social structural influences.[4][5] However, very few studies have examined the determinants of non-adherence among people with a rare disease, defined by the European Union as conditions that affect fewer than 5

in 10 000 people in the general population.[6] Rare diseases in totality are however not so rare, with over 6000 rare diseases identified. It is also estimated that 7% of the population will be affected by a rare disease at some point in their lives.[6]

This study focuses on adherence to recommended lifestyle changes among patients with xeroderma pigmentosum (XP), a rare disease caused by mutations in any of eight genes, resulting in defective repair of ultraviolet radiation (UVR) induced DNA damage.[7] The incidence is 2.3 per million live births in Western Europe.[8] Patients may develop skin cancers from childhood, with an estimated 2000-fold and 10 000-fold increased risk of melanoma and non-melanoma skin cancers, respectively, together with increased risks of ocular problems and neurodegeneration.[9] About half the patients were 'burners' who suffer abnormal and severe sunburn reactions and blistering as measured by an XP sunburn score and are at the highest risk of neurological degeneration. 'Non-burners' have an entirely normal sunburn response but have the highest risks of pigmentation changes, skin cancers at an early age and ocular problems.[10]

There is currently no curative treatment for XP. Improving life expectancy and reducing risks of skin cancers and eye disease therefore depends on rigorous long-term UVR protection. This involves regular use of sunscreen (Sun protection factor (SPF) 50+), covering arms and legs with layers of clothing and wearing gloves, a UVR protective visor or other head/face covering, UVR protective glasses and attaching UVR protective film to windows.[11]

Research to date has shown that sun protective knowledge is high among general populations[12] and among high-risk groups[13 14] but has not adequately explained low adherence. Furthermore, no research has focused specifically on people with XP, although expert clinical opinion suggests these patients may vary widely in the degree to which they photoprotect (H Fassihi, personal communication 2016).

The current study forms one component of a 5-year National Institute for Health Research (NIHR) funded programme of research that aims to develop and evaluate interventions to increase adherence to photoprotection and thus reduce measured UVR exposure among people with XP.[15] The programme is underpinned by a systematic theoretically based framework to guide behaviour change interventions.[16] The first mixed-methods phase therefore aimed to achieve a comprehensive understanding of photoprotection behaviour to inform the development of individualised photoprotection interventions.[15] The quantitative studies comprised a cross-sectional survey to assess potential clinical, sociodemographic and behavioural correlates of photoprotection behaviour and a number of one (N-of-1) study using daily diaries to record photoprotection activities and ratings of potential psychosocial predictors. These were complemented by this more in-depth qualitative study to explain photoprotection behaviour.

## AIMS

To elicit individuals' personal accounts of their photoprotection behaviour and identify the range of factors that influence their behaviour.

## METHODS
### Design

Inductive qualitative study based on semistructured interviews that were analysed following a framework approach.[17] The Standards for the Reporting of Qualitative Research guidelines were followed.[18]

### Patient and public involvement

Six members of the patient and public involvement (PPI) group were recruited. However, two parents dropped out as their child was having severe health problems. The remaining four PPI members (one patient, two parents, one teacher) participated in two discussion sessions to consider the qualitative findings. One parent was a co-investigator and as part of the research team attended meetings throughout the research from the initial development of the application.

All study participants were sent a brief newsletter that provided study feedback. They were also invited to a thank you event that included informal discussion of the results. Further dissemination will occur following the trial phase of the research.

### Recruitment

The study sample was recruited from the national specialist XP service in London that is part of the National Health Service with universal access based on medical need. Eligible adults were identified from the register of clinic attenders and satisfied the following criteria: aged 16 years and over, confirmed diagnosis of XP (reduced unscheduled DNA repair in fibroblast DNA repair assay), no neurodegeneration, adequate English to participate in an interview and had not opted out of research.

A research nurse, who was employed by the research programme and had no clinical contact with patients, organised study recruitment. She sent study information to the 38 eligible patients and 2 weeks later followed up with a phone call to enquire whether they were interested in participating. Seven could not be contacted and six declined due to time or not wishing to talk about XP. These six patients comprised three men and three women with a spread of ages. Their clinic notes also identified varying photoprotection behaviours thus identifying their similarity to the interviewed sample.

### Data collection

Twenty-five one-to-one interviews were conducted. They were held in a private room in the patient's home with the exception of one phone interview for logistical reasons. Interviews were carried out between early February and June 2016 by a research fellow (JWal) and research assistant (RA) who are both trained in health

psychology and are experienced interviewers but had not previously studied XP. Participants were aware that they were non-clinical researchers employed at King's College London to conduct the research. An interviewer (female) visited the patient's home with the research nurse who had made the initial contact with participants to provide continuity. She introduced the interviewer and obtained informed consent but was not present for the interview and when required looked after participants' children.

Interviews took the form of a guided conversation structured by a topic guide (online supplementary table 1). They began with a brief reminder of the aims of the study and then focused on aspects of the patients' personal story to identify individuals' early experience of XP. This was followed by eliciting their current views and experiences of photoprotection, the psychosocial impacts of photoprotection and support by family and friends. Topics were discussed in any order according to the flow of the conversation and responses probed and clarified as necessary. Interviews were audio-recorded with permission and mainly took 30–60 min. Participants were generally willing to provide detailed accounts of their photoprotection behaviour. This is likely to have been facilitated by the home environment and their positive views of both the clinic and the aims of the research. However reflective notes made following the interview identified a few participants becoming upset in thinking about XP. Some also preferred not to discuss earlier stages and only to focus on the present. However, they were all happy to continue the interview.

Data saturation was achieved with the 25 interviews with no new beliefs or explanations identified in the final interviews.

## Analysis

This occurred alongside data collection and was based on a framework approach.[17] This goes beyond a thematic analysis and involves methods that are geared towards producing practice-oriented findings and was therefore appropriate for a study aiming to inform behavioural change interventions.[19]

Tapes were transcribed verbatim, checked for accuracy and corrected transcripts entered into QSR NVivo V.11 which is a qualitative data analysis package designed for working with text or multimedia information where in-depth analysis is required (https://www.qsrinternational.com/nvivo/what-is-nvivo). Transcripts and field notes were initially read to identify key themes. Coding was then undertaken with the initial coding framework adapted as new items arose. One author (RA) had a major role in coding (ie,' indexing') with two other authors (JWal, MM) repeating and reviewing the coding which was discussed to achieve consensus. Analysis involved comparisons within and across cases assisted by 'charting' which involves summarising data by thematic content for each case to identify relationships between codes.[19] These charts were further examined through a process of 'mapping and interpretation'

leading to emerging ideas and relationships. Preliminary behavioural categories were tested through further analysis of transcripts to identify the fit for all cases with modifications introduced as required. This iterative process finally led to a threefold categorisation based on perceptions of risks of UVR and the psychosocial impacts of photoprotection.

## Credibility and triangulation

The credibility of emergent categories in terms of their congruence with reality forms a key aspect of validity in qualitative research.[20] In this case credibility was informed by the PPI group who participated in two sessions to discuss the qualitative findings and in two clinical discussion groups held with a total of 11 clinical specialists who undertake the patients' annual check to identify the congruence of our identified patient categories with their clinical experience. The level of adherence with photoprotection of our three patient categories was also examined based on prior clinic reviews by specialist nurses using a 13-item measure.[21]

## RESULTS
### Characteristics of participants

Twenty-five adults were interviewed. Participants were aged 16–63 years and 18 lived outside London. Seventeen were male and 8 female reflecting the adult clinic case load (26 male and 17 female). Eight were European (mainly white British), eight of Pakistani ethnicity and seven mainly of Middle Eastern, Indian or Bangladeshi background, reflecting the relatively high prevalence of XP in Pakistan and the Middle East.[7]

Fifteen participants were defined as 'non-burners' (ie, normal sunburn response) and 10 as 'burners' (ie, abnormal sunburn response to minimal sun exposure).[10] Those categorised as 'burners' had all begun some form of photoprotection in childhood following recognition that they burned easily, whereas half the 'non-burners' had begun photoprotection after 10 years of age. Self-reported clinical diagnosis occurred more than 5 years ago for all but five participants and all currently attended the specialist clinic annually. The 'burners' had experienced reddening and blistering, pigmentation changes, particularly freckles and benign lesions. The 'non-burners' had often experienced some pigmentation changes and nearly all had experienced removal of one or more cancerous lesions.

### Knowledge of risks and photoprotection

Participants were all aware that XP is a long-term condition with risks of skin damage and lesions. They were also aware of requirements for photoprotection and explained that their knowledge had increased over time through greater experience of the condition and attending the specialist clinic. For example, some participants initially thought only the face is affected by UV exposure and were therefore surprised to require lesions removed from

---

**Box 1    Behavioural responses to photoprotection among people with xeroderma pigmentosum (XP)**

**Dominated by demands of photoprotection**
The major driver for rigorous photoprotection was a high fear of ultraviolet radiation (UVR)-related risks. Although concerned about the impacts of photoprotection on activities and experiencing emotional distress, these did not lead to reduced photoprotection.

**Resistant to photoprotection**
The major driver was the desire to live a normal life which conflicted with rigorous photoprotection.
For a group of predominately younger adults their desire to distance themselves from a stigmatised identity and be accepted as normal by their peer group outweighed concerns about the clinical risks of UVR. This led to 'strong resistance' in terms of concealing their condition by undertaking fairly limited photoprotection and often not disclosing their condition to others.
For a group of mainly older adults the requirements for photoprotection had no significant impact on self-identity or disclosure but they questioned the personal risks of UVR, its perceived efficacy and the priority given to other things in life. This led to 'occasional resistance' with gaps in their photoprotection, especially going out for short periods without sunscreen.

**Integrated photoprotection**
UVR protection was integrated into normal life and not perceived as disruptive to activities or an emotional burden. Some achieved this by largely staying indoors which they were used to from childhood and was their preferred life style. Others lived a more usual life and were assisted in this by the cooperation of friends/family and among older people by the changing values, lifestyles and expectations of older age. Many participants, although confident in their photoprotection, also acknowledged scope for improving their routines.

---

their neck, arms or legs. As participant 28 (female, aged 28 years) explained, '*.so my neck, so then I'll wear a hat and I'll wear sun cream more often, and I'll wear my sunglasses. Then there's bits on your arms, so I'll start wearing long sleeve t-shirts. And then this happened on my leg.but now I'm like no, I'm going to buy leggings and I'm going to wear leggings with long sleeve tops.'*

All participants engaged in some photoprotection but, when asked why they did not undertake particular aspects, often found this difficult to answer. However, further discussion identified a range of positive and negative factors that influenced how they interpreted and balanced medical and psychosocial priorities. This led to identifying three main types of response to photoprotection: '*dominated*' by photoprotection, '*resistant*' to photoprotection and '*integrated*' photoprotection (box 1). See online supplementary table 2 for participants' age/sex and behavioural response category.

## Responses to photoprotection
### Dominated by demands of photoprotection
This group comprised four men (one 'burner' and three 'non-burners') aged between 21 years and 28 years who all described a very high level of photoprotection. Their major motivation for undertaking rigorous photoprotection was

their perception of the harmful effects of UVR. The three non-burners feared cancerous lesions and worried about a reduced length of life, whereas the major worry for the one 'burner' (participant 15 aged 22 years) was having very painful and unpleasant sunburn reactions. However, for all participants an immediate concern was the effects of UVR on their appearance due to freckling and other pigmentation changes including facial scars following surgery. For example, participant 14, a non-burner (male, aged 21 years), explained, '*If you take it purely objectively, I don't think I'm as unfortunate as some skin patients are (effects on appearance). But you still worry about it. Yeah, that's a main worry.'*

People 'dominated' by photoprotection had begun sun protection in childhood but described adjusting better to it as they got older and often did not want to think back to earlier stages, describing this as '*too long ago'*. However, participant 7 (male, aged 27 years) explained that at school and college he never used his sunscreen '*because I wanted to feel normal, that's why. I didn't want any questions, or any remarks from friends or people that I didn't know, and I did not want to look abnormal to anyone else.'* This changed when he had his first surgery to remove a basal cell carcinoma which he described as a '*reality check'* and '*wake-up call'* as '*I was putting myself in harms' way for no reason…just destroying my own skin.'* From that time he gave considerable emphasis to the risks of UVR and the need for rigorous photoprotection.

All four respondents described having UVR protective film on the windows of their home and car and used a UVR meter to assess risks and identify appropriate environments and locations. They also described well-established behavioural routines for photoprotection. This involved applying sunscreen whatever the season and wearing two or three layers of protective clothing, gloves and a hat or cap with a hoodie or a visor when outside during the day. A visor provides full cover of the face and head and is usually only worn by children but was worn by three people categorised as 'dominated'. They described a visor as uncomfortable to wear, bulky to carry around, and attracting stares and comments. However, users regarded this as necessary to achieve sufficient protection and regarded wearing a visor as part of '*going out properly protected'*. The one participant in the 'dominated' group who did not wear a visor but otherwise engaged in rigorous photoprotection described a visor as 'suffocating' and as having negative psychosocial impacts, '*.totally make me feel abnormal. Just wearing a hat, cap and glasses all the time, I've gotten people questioning like mad, so imagine wearing a visor.'* (participant 7, male, 27 years)

Participants 'dominated' by photoprotection explained how their practices had shifted from a deliberative action into a routine or 'habit' which they now *'do without thinking, it becomes part of your daily routine'.* As one participant observed, '*I think I'm motivated by keeping myself safe and I don't want to develop any skin cancers, which is a problem for my condition. But really, rather than motivation it's something I do just because it's a deeply ingrained in me*

*and because I've been doing it forever.'* (participant 13, male, aged 28 years)

This high level of photoprotection carried considerable personal costs. These included restrictions on choice of job with risks of exposure to UVR forming an important consideration. As one participant (no.14, male, aged 21 years) explained, *'It's affected the kind of jobs that I can look for. I only choose something that's inside in a safe environment and I really can't go for anything else.'* He now works as a shop assistant in a large supermarket. Other participants were also employed in safe environments, as a barber, in the family takeaway and as a health scientist, rather than doing their job of choice.

Participants also described missing out on social events because they did not want to wear UVR protection or did not regard it safe to go outside even with full protection. For example, participant 13 (male, aged 28 years) explained, *'I make excuses about not going out with friends and prefer to be indoors between 11 and 4 when the sun is at its highest.'*

If going out during the daytime was unavoidable they undertook considerable preparation. As participant 22 (male, aged 26 years) explained, *'the only times I go out during the day is if I'm going from a car to a building or building to car, or if it's completely unavoidable….'*

This decision in turn cues his usual routine that involves *'suncream, visor, everything'*. He described this as taking him *'……about half an hour just to get ready because I've got to apply my sun cream, then I've got to be sure I haven't missed any areas. I also apply lip balm which is UV protected. Then once I've applied my sunscreen I apply my camouflage (to hide scars). I apply two camouflages.'* He also explained, *'If I've got to be somewhere for 5pm I'll have to leave at 4.30pm which means I've got to cover up between (workplace) and the tube station. So, I need the visor that day.'*

This group referred to the psychological costs of a high level of photoprotection in terms of being viewed as different or not 'normal'. They described no longer being upset by stares and questions about their clothing and if people continued to ask questions or make comments, they now felt more confident and able to *'put them in their place.'* However, they still sometimes avoided situations that might lead to *'snide remarks',* such as taking a taxi rather than a bus when school children were travelling.

Despite considerable support from family and friends, this group described feeling emotionally distressed about the demands of photoprotection and restrictions on their activities. For two participants the impacts were quite severe, for example, participant 7 (male, aged 27 years) explained, *'whatever I do XP is on my mind first'* and described XP as *'affecting me emotionally quite a bit as well.,It's just you get obsessed and get that depressing feeling. You can't do much in life.'*

Similarly, participant 22 (male, aged 26 years) described *'feeling down'* about having XP and its effects on his life and got *'bouts of depression'* about three or four times a year. However, these various adverse effects did not disrupt

their high level of photoprotection given their fears of the potential clinical risks.

## Resistant to photoprotection

This group comprised 11 participants, aged 20–55 years, with 5 'burners' and 6 'non-burners'. All accepted their diagnosis of XP and were aware of the risks of UVR and photoprotection practices. They therefore had the capability to photoprotect but acknowledged a lack of motivation with other priorities.

We identified five people (two men and three women, aged 21–37 years) as 'strong resistors' as they described very limited photoprotection that largely reflected their desire to distance themselves from XP and be accepted as normal by others. For example, participant 24 (female, aged 28 years) had experienced several surgeries to remove skin cancers and benign lesions and was particularly concerned about surgery to her face. Nevertheless, she described 'strong resistance' to photoprotection and often joined her friends for activities in the sun, did not consistently reapply sunscreen and sometimes went out wearing shorts or summer tops, because as she explained, *'I just want to be seen like everybody else'* and *'I think you can just let these things (photoprotection) become you and I don't want to be about XP. XP is a tiny part of me.'* She therefore sometimes thinks, *'…do you know what, I'm just going to get on with my life.'*

She excused her behaviour at the interview by saying that when she was out she 'shade hopped' and also gained reassurance from knowing that the clinic would identify and manage any problems.

Concerns about feeling different and viewed by others as possessing an unacceptable difference was common in this group although their particular circumstances differed. For example, an Asian participant (participant 6, male, aged 20 years) described freckles as highly unusual and abnormal on a person with Asian skin and reported that comments and questions by the Asian community made him feel *'like an outsider'* which corresponds with the notion of 'felt stigma' or feelings of shame associated with an unacceptable difference.[22] This response contributed to his stress and loneliness and to a reluctance to increase the obviousness of his condition through rigorous photoprotection.

This subgroup of 'strong resistors' therefore described tensions between their desire to be seen and participate as normal, and worries about the effects of UVR without sufficient protection. For example, participant 18 (female, aged 27 years) worries about cancer but often feels *'a bit left out'* as she can't go outside at lunch with colleagues. She also acknowledged that although she sometimes wears long sleeves in summer, she does not usually wear a hat or reapply sunscreen which she regarded as greasy and staining. She explained that *'some days I don't bother with cream or covering up…Sometimes I think screw it, just go out. Then I think, oh, sugar,….But there's consequences, just like oh my god.'* Similarly participant 12 (male, aged 26 years) explained that he worries about life expectancy and

appreciates the advice and support given by the clinic but still goes out and socialises in the summer '…*just to make myself feel better really, That's about it.*'.

Concerns about being perceived and responded to in terms of XP led to concealing the condition by reduced photoprotection, the use of a tinted foundation with a lower SPF and reluctance to disclose their condition to others. Reminders to photoprotect by family members were also often described as unhelpful as participant 12 (male, aged 21 years) explained, '*I just think it's a constant reminder that I'm different. So that's why I don't really like them (reminders).*'.

The other six people formed a subgroup of 'occasional resistors'. They were all men and mainly in their mid-30s to mid-50s. They were not bothered about people knowing about their XP but had gaps in their photoprotection routines, often going outside for short periods without sunscreen. This appeared to partly reflect perceptions of low personal risk if they had lived 20 years or 30 years without major problems. There was also a view that life is uncertain and that exposure to UVR is just one of many risks they might encounter and so not a priority. They therefore preferred to '*live for today*' *and* have '*a happy life despite XP*'. As participant 20 (male, aged 55 years) explained, '*I don't automatically put on sunscreen everyday even though I know I must.…there's no downsides to sunscreen, so it's just time or not thinking about it or being too busy, or thinking no I can't be bothered.*' He later explained, '*I would rather do things that give me pleasure in life rather than get overly anxious about the ransom to the possible outcomes of not taking enough care.*'

Some people referred to specific personal experiences as shaping these views, such as a close relative dying unexpectedly at a young age. Other reasons given for resisting rigorous photoprotection was their questioning the effectiveness of sunscreen or of photoprotection, generally, especially if they had experienced lesions on parts of the body normally covered.

All participants in the 'resistant' category thus emphasised the importance of leading a normal life but differed in whether the major influence on their behaviours arose from feelings of unacceptable differentness, or from perceptions of personal UVR risk being outweighed by other life priorities and risks.

## Integrated photoprotection

This category comprised five men and five women who represented the full age range from 16 years to 63 years, with four 'burners' and six 'non-burners'. All currently accepted the need to undertake photoprotection, although a few people described struggling initially as the demands of photoprotection made them feel like a 'sick' person. However, over time these practices had become '*automatic*', '*habitual*' and '*just second nature*'. Photoprotection was therefore not currently viewed as a major practical or emotional burden leading to a 'non-normal' life and illness identity but described as '*just a part of life … It's just habit now, a routine, what I do.…It's just second nature*

*I think for me*' (participant 19, male, aged 18 years). Similarly photoprotection was referred to as '*.no hardship to do because I just do it as part of my daily routine*' (participant 16, female, aged 63 years). These participants generally felt comfortable and confident in their photoprotection routines and currently did not worry much about the risks of UVR.

This accommodation to photoprotection was described as occurring in one of two ways. A group of younger people diagnosed in childhood had adapted by spending most of their time indoors and now preferred this. For example, participant 19 (male, 18 years) was diagnosed when 9 years old but began to protect much earlier. Currently his routine was to set off for work at 6:00 when dark and his father collected him later by car. Otherwise he was mainly in the house and preferred being indoors where he played with the computer. Participant 1 (female, aged 45 years) also described her need to keep out of the sun as fitting quite well with her life because she was at home most of the day looking after children.

Others emphasised how they accommodated the requirements of photoprotection without withdrawing indoors. For younger people this was often facilitated by friends or family members who accepted their need to avoid UVR and adjusted activities to ensure that they were not made to feel different or excluded.

'*….especially when we're going out and there's a place that's not safe and then they all understand we probably need to move somewhere else and not do what everyone wants to do, which is to enjoy the sunlight.*' (participant 9, male, aged 38 years)

Family members often organised activities when the sun was low, encouraged indoor sports and going on holiday in the UK rather than to hotter countries. Similarly, partners might adapt their roles, as participant 10 (male, aged 39 years) explained,

'*I'm the one who would normally be staying in the car* (with window film*).and the missus would actually go into the shops, into the street to find whatever we need to get.*', '*My wife tends to do the outdoor activities with the kids, and I do the indoors.*'

For older people their accommodation of photoprotection had often become easier through their longstanding experience of XP. As participant 16 (female, aged 63 years) commented, '*Because I'm this great age now, I'm used to it.…it's just part of my daily routine now.…the only time it impacts is if I'm invited to an outdoor event and either I don't go or wear a hat and stick to the shade.…but that's the only time it impacts you. But I'm used to it now*' (laughter).

For older people, the personal social costs of photoprotection were also often facilitated by changes in what they expected and wished to do, as participant 21 (male, aged 62 years) explained, '*When younger I wanted to sunbathe and go on holidays with friends to hot countries but as the years went by I got used to it. Now it doesn't bother me.*'

Older people who had integrated photoprotection in their lives also described rejecting any stigma associated with photoprotection, with any negative views and comments described as '*other people's problem, not my problem*'.

People who had integrated photoprotection into their lives described feeling confident that their routines provided adequate personal protection. However they sometimes acknowledged scope for improvement, including the need to apply sunscreen more regularly and give more emphasis to UVR protection on cloudy days. They may also have occasionally not been aware that their photoprotection practices were not sufficiently rigorous as these had become well established habits they no longer thought about.

### Credibility and triangulation

Findings of different responses to photoprotection advice were presented and discussed with the four active members of the PPI team. They endorsed the findings with a minor revision regarding a subgroup that was adopted.

The two discussion groups with clinical specialists indicated that they regarded the three emergent categories as fitting with their clinical experience. They also viewed the data as helpful in increasing their understanding of the range of psychosocial meanings and contextual influences on photoprotection behaviour.

Patient data for a 13-item measure of photoprotection that is routinely used by clinic nurses to assess adherence with photoprotection[21] was examined following completion of data analysis so as not to influence how we interpreted the interviews and categorised participants. This recorded very high photoprotection behaviour scores for the 'dominated' group, whereas the 'resistors' had the lowest mean score with all scoring 7 or less apart from one participant (participant 8, male, 55 years) whose score was higher as a result of having dermagard on windows at home and in the car, although behavioural aspects were in line with other resistors. Overall the 'integrated' group had a moderate score, although half had a low score indicating that their routines were often not sufficiently rigorous (table 1).

**Table 1** Clinical assessment of photoprotection protection score * for three behavioural categories

| Behavioural categories | Range† | Mean protection score (SD) | Protection score rating* |
|---|---|---|---|
| 'Dominated' | 16–20 | 18.25 (2.06) | High |
| 'Resistant' | 3–14 | 6.18 (3.25) | Low |
| 'Integrating' | 7–15 | 9.90 (3.00) | Moderate |

High score=15–20, moderate score=9–14, low score=0–8.
The score does not take account of ultraviolet radiation protection with heavy curtains or blinds, with only dermagard window film at home receiving a score (100% day protected=6; 50% day=3). This reduced the score for people who spent most of their time indoors with curtains or blinds pulled.
*Based on Ultraviolet Radiation Protection Measure applied by specialist nurses (Henshaw and Turner, 2016).[21]
†Scores out of 20 based on 13 items covering five areas: (1) Window film. (2) Sunscreen. (3) Face visor or glasses/ultraviolet radiation eye protection/face buff/hat. (4) Hand protection. (5) Arms/legs covered.

## DISCUSSION

Three broad groups of responses to photoprotection were identified. One group, termed 'integrated' represent what is often regarded as the ideal form of adjustment among people with a chronic condition and represents the final stage of the self-management pathway in chronic illness that is often referred to as 'normalisation'. This term sometimes emphasises psychological aspects in terms of bracketing off the impact of illness so that its effects on the person's identity are relatively slight,[23] or may refer to the incorporating of illness or treatment regimens into what is viewed as a normal life.[24] Patients categorised as 'integrated' displayed both features of normalisation; they no longer felt concerned about possible negative responses and their photoprotection had moved from a deliberative to an automatic and habitual process and was now a routine part of their life. This successful adaptation was accompanied by photoprotection practices they felt comfortable and confident with, although participants' accounts and adherence scores indicated that they did not necessarily achieve the ideal in terms of rigorous photoprotection and had a more moderate level of photoprotection. In contrast those 'dominated' by the requirements of photoprotection achieved high levels of UVR protection but at a high cost in terms of activity restrictions and emotional distress and resulted in, what Sanderson et al in their study of rheumatoid arthritis referred to as, a 'disrupted normality'.[25] However, for XP this disruption largely occurred through the visibility and demands of protective measures rather than through overwhelming pain or other physical symptoms. A third group of 'resistors' did not aim to normalise photoprotection, with the 'strong resistors' engaging in, what Sanderson et al[25] referred to as, 'struggling for normality' at all costs, possibly trading short-term social normality for increased clinical risks and reduced normality in the longer term. This reflects descriptions of 'denial' of an illness identity and responses to stigma in relation to a range of chronic conditions, particularly among adolescents and young adults.[26–28] In contrast, a few 'occasional resistors' partly normalised photoprotection in their life and described some established photoprotection routines but also questioned the personal priority of photoprotection and acknowledged sometimes going outside for short periods without bothering to apply sunscreen.

This study thus identifies some similarity in the responses of people with XP and those identified for other chronic diseases. It is also notable that the group 'dominated' by photoprotection and who had the highest adherence scores experienced considerable psychosocial disruption, whereas those who had achieved greater psychosocial normalisation disclosed that they did not necessarily do all they could in terms of photoprotection and had lower photoprotection scores. It was also clear that some apparently irrational behaviour, particularly among the 'resistant' group, was perceived as rational by these participants given the importance they assigned to having a normal life and identity.

The differing behavioural responses to XP correspond with Horne et al's 'Necessity-Concerns' framework[29] developed to explain medicines use, with individuals' photoprotection

practices reflecting their perceptions and balancing of the perceived necessity of photoprotection to reduce UVR risks with concerns about the impacts of photoprotection on activities, feelings of stigma and emotional distress. For the group 'dominated' by photoprotection their belief in the necessity of photoprotection clearly outweighed the emotional distress and activity restrictions they experienced, whereas these psychosocial concerns were of particular significance and influences on the behaviours of the group of 'strong resistors'. This may have been influenced by their age and life stage, personality and general orientation to risk taking, although these aspects were not investigated. More broadly, the data indicate that changes in either perceived necessity of photoprotection or concerns about the impacts of photoprotection could lead to shifts in behaviour.

The study interviews focused on individuals' current perceptions and behaviours in terms of accommodating and coping with requirements for photoprotection. However, a few participants described transitions. One transition was from 'resisting' to 'dominated' that was triggered by greater perceived necessity through their experience of a skin cancer. There was also evidence of a shift towards 'integration' which appeared to be influenced by increasing age, greater resilience and changing social expectations. However triggers and processes of change between different behavioural responses require detailed study with a longitudinal design. There are also questions of the influence of early childhood experiences, including feelings of stigma and the emphasis given by families to either facilitating their child's participation in normal activities or engaging in a UVR avoidant strategy involving restricted time outside.[30][31] These early experiences may thus have shaped adults lifestyle choices and their attitude to health risk which is known to have an important influence on preventive health risk behaviour and treatment preferences.[32] Researching these issues would contribute to what Moss-Morris[33] noted is a need for a more adequate understanding and theory of adaptation to chronic illness, while Leventhal et al[34] recommended that greater attention should be given to the dynamic processes involved in coping with chronic illness and key transitions, including transitions in relation to adherence.

### Study limitations

The study of a rare disease involves a number of challenges.[35] These include the necessarily limited study population that requires particular attention to ensuring confidentiality and anonymity and risks of considerable respondent burden. For example, it was not feasible to undertake further follow-up interviews to explore emerging issues in greater depth or undertake respondent validation as this patient sample was also participating in other phase I studies and will be invited to participate in the subsequent trial. A further limitation is that clinic attenders with neurodegeneration, which occurs in around a quarter of people with XP,[9] were excluded as they are likely to face considerably different issues to people without neurodegeneration and thus require a separate study.

### Informing behavioural interventions

This qualitative study complements the quantitative aspects of the phase I programme and provides greater depth and richness of data to explain the 'why', 'what' and 'how' questions. For example, why participants went out in the sun without full protection, what they felt were the impacts of photoprotection on their lives, whether and what they disclosed to friends, and how they responded to questions about their appearance.

These findings will inform the design of the intervention in a number of ways. This includes identifying people 'dominated' by photoprotection who already have a very high level of adherence and for whom the intervention would therefore not be appropriate. However, their accounts of emotional distress identified the potential risks associated with high levels of photoprotection and indicates that emotional well-being will need to be probed in the intervention sessions and responded to for individuals as appropriate.

The study identified people 'who were resistant' to photoprotection as having few established routines and most likely to experience a constellation of barriers to photoprotection, while those categorised as 'integrated' shared some of these barriers. This indicates that the intervention should include both generic components and components personalised to the specific barriers relevant to each person with the aim of changing individuals' necessity-concerns balance. Generic components might include developing positive 'habits' that refer to a situation in which a stimulus (eg, going outside) generates an impulse to act as a result of a learnt stimulus response, with a positive cue-action link leading to protective behaviours.[36] Habit formation training could involve action planning and the use of environmental cues and prompts (eg, intervention text message) to instigate the specific photoprotection behaviour. The components to address specific needs and barriers could include strategies to enhance acceptance and willingness to photoprotect as well as strategies to reduce felt stigma by managing appearance concerns and to facilitate disclosure. Patients' accounts of their lived experiences provided by the qualitative interviews could also be included in written materials to support the one-to-one sessions and overcome personal barriers to photoprotection through drawing on the direct experience of other patients.

### CONCLUSIONS

This study goes beyond the identification of themes to examine the complexity and patterning of responses, including the influence of patients' illness experience, priorities and circumstances of their lifeworld[37] on their photoprotection behaviour, and contributes to an understanding of adaptation and normalisation in chronic illness. The data will also inform the content and design of individualised interventions to address both motivational and volitional barriers to photoprotection of people with XP and may also be of value for other conditions with high risks of skin cancer.

**Acknowledgements**  The authors thank Lesley Foster who was instrumental in recruitment and facilitating the interviews. The authors also thank the clinical specialist XP team at St Thomas' Hospital for helpful discussions and specialist nurses Tanya Henshaw and Sally Turner for adherence data. The authors also thank the PPI panel (Cathy Coleman, Ben Fowler, Ros Tobin, Sandra Webb) for inputs and are most grateful to the study participants for the detailed accounts of their experiences and photoprotection practices.

**Contributors**  MM directed the study and was primarily responsible for study design, analysis and writing the paper. RA and JWal conducted interviews, checked transcripts and were involved in the development of project materials and interpretation of findings. RA had a major role in coding with MM and JWal contributing. JWal gained ethical approval. Both RS and JWei as the programme leads were involved in discussions throughout the research and contributed to study design. All authors critically reviewed and contributed to the final paper.

**Funding**  This article presents independent research funded by the National Institute for Health Research (NIHR) under its Programme Grants for Applied Research scheme (RP-PG-1212-20009). The views expressed are those of the authors and not necessarily those of the NHS, the NIHR or the Department of Health. NIHR had no involvement in study design, research process or manuscript preparation.

**Competing interests**  None declared.

**Patient consent**  Obtained.

**Ethics approval**  London-Camden and Kings Cross Research Ethics Committee (REC number 15/LO/1395).

**Provenance and peer review**  Not commissioned; externally peer reviewed.

**Data sharing statement**  Due to concerns about anonymity in such a small rare disease group our research team has decided not to put raw data in a repository.

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
