## [Reviewer comments · BMJ Open]

ARTICLE DETAILS

TITLE (PROVISIONAL)	The influence of perceived medical risks and psychosocial concerns on photoprotection behaviours among adults with Xeroderma Pigmentosum: A qualitative interview study in the UK
AUTHORS	Morgan, Myfanwy; Anderson, Rebecca; Walburn, Jessica; Weinman, John; Sarkany, Robert

VERSION 1 – REVIEW

REVIEWER	Anthony Bewley Barts Health NHS Trust, UK
REVIEW RETURNED	09-Jun-2018

GENERAL COMMENTS	This is a well designed and executed study.
---

REVIEWER	Kenneth Kraemer, M.D. and Deborah Tamura, R.N. National Cancer Institute, Bethesda, MD USA
REVIEW RETURNED	21-Jun-2018

GENERAL COMMENTS	Influence of perceived medical risks and psychosocial concerns on photoprotection behaviours among adults with Xeroderma Pigmentosum: Qualitative interview study in the UK. Myfanwy Morgan, Rebecca Anderson, Jessica Walburn, John Weinman, Robert Sarkany. Comments for the Authors This is a very interesting study assessing the use of photoprotection by a group of XP patients attending an XP specialty clinic in Great Britain. This qualitative study was accomplished using structured interviews with 25 adult XP patients in their homes. They describe 3 groups of patients with respect to sun protective behavior: Dominant; Resistant; and Integrated. The results of the study will serve as a baseline for developing interventions to teach UV protection; the findings of this study will also be useful for clinicians currently working with XP patients in determining UV protection attitudes. 1. Abstract, Strengths and Discussion: Please delete claims of being “first” – this is nearly impossible to prove and is not relevant 2. Introduction: One of the limiting aspects of the study not mentioned by the authors is the universal availability of health care. There are members of different ethnic groups in the study, but they all have access to the same health care system. The costs of the UK clinic specialty clinic and follow-up health care costs are covered by the National Health Service at little to no out of pocket cost to patients. XP patients in countries where universal health care (and for that matter any health care) is not available may experience different attitudes toward XP and UV protection. The findings in this study may not be generalizable to countries with less reliable health care access or developing countries.
---

	3. Page 6 Line 40 It would be helpful to provide a short paragraph on the patients who opted out of the study. Were there similarities between those patients? 4. Page 6 Line 55: Please clarify if the research nurse was from the XP specialty clinic or from the author's group. 5. Page 7 Line 46: A brief description of the QSR NVivo11 software would be helpful. 6. Page 8. Table 1. Ordering the table by patient study number doesn't provide useful information. Please either order it either by age or by UV protection group. 7. Page 12 Line 5: The 'Resistant to Photoprotection' group also appear to be more 'risk takers'. Was that assessed? 8. Page 31: Line 14: The 'Integrated photoprotection' group appears to be the most adapted to living with XP photoprotection. There is a concept called Normalization which has been researched with several papers published in the nursing literature. The construct explores a change process experienced by individuals adapting behaviors to care related changes brought on by the demands of chronic disease. It may be helpful to examine the review paper by Deatrack et al IMAGE JOURNAL OF NURSING SCHOLARSHIP 1999; 31 :3, 209-214 or other papers on normalization. Minor comments:  1. Abstract: define PPI group earlier than on page 8. 2. Page 6 line 54 – XXx college? 3. Page 8 Line 56: The sentence is misleading and should be corrected. It seems to say that 'non burners' only get cancers – they also get pigmentation often before the 'burners'. 4. Page 11 Line 4: Thar – is a misspelled word. 5. What is reference 16? Team protocol paper? Also reference 26 – "Team paper"?
--	---

VERSION 1 – AUTHOR RESPONSE

Reviewer 1: Thank you for your very positive assessment.

Reviewer 2: Your helpful comments have all been addressed.

1. Abstract – deleted claim to being 'first'
2. Introduction – we have included a statement explaining that the context of study was a health system with universal access in the Recruitment section (p. 6) and discussed this in Strengths and Limitations.
3. We have described the six eligible patients who opted out of the study as having similar characteristics in terms of a mix of men and women, varying ages and responses to photoprotection as the interview sample (p.7)
4. We have explained that the research nurse was employed through the research programme and was not on the clinic staff or having clinical contact with patients (p. 6)
5. A brief description to QSR NVivo 11 is now provided and a website cited (p.7)
6. We have now removed Table 1 as BMJ Open only allows a maximum of two indirect identifiers (e.g. age and sex) in a table to ensure anonymity.
7. We did not assess more general risk taking among the resistant group but now comment on its potential significance (p.18) and provide a reference (33).
8. Thank you for your comment about normalisation and for providing a reference. We have included this and also revised para1 in the Discussion (p. 16) to address normalisation and also refer to this on p.17.

Minor points:

1. The PPI group is now defined earlier (pp. 5-6)
2. Have now inserted 'King's' College which was omitted to achieve anonymity (p.6).
3. We have changed the sentence describing 'non-burners' (p. 8 para2)

4. Misspelled word Thar (p. 11 line 4) – sorry I could not find this.
5. References to ‘team papers’ nos. 16 & 26 (now 31) given for purposes of anonymity and have now put the full references to these papers.

VERSION 2 – REVIEW

REVIEWER	Kenneth Kraemer and Deborah Tamura National Cancer Institute, NIH, Bethesda, MD USA
REVIEW RETURNED	14-Sep-2018
GENERAL COMMENTS	Good revision!